# Chirality coupling in topological magnetic textures with multiple magnetochiral parameters

Oleksii M. Volkov ◉[1] ✉, Daniel Wolf ◉[2] ✉, Oleksandr V. Pylypovskyi ◉[1,3], Attila Kákay ◉[1], Denis D. Sheka ◉[4], Bernd Büchner[2,5,6], Jürgen Fassbender ◉[1], Axel Lubk ◉[2,5,6] & Denys Makarov ◉[1] ✉

Chiral effects originate from the lack of inversion symmetry within the lattice unit cell or sample's shape. Being mapped onto magnetic ordering, chirality enables topologically non-trivial textures with a given handedness. Here, we demonstrate the existence of a static 3D texture characterized by two magnetochiral parameters being magnetic helicity of the vortex and geometrical chirality of the core string itself in geometrically curved asymmetric permalloy cap with a size of 80 nm and a vortex ground state. We experimentally validate the nonlocal chiral symmetry breaking effect in this object, which leads to the geometric deformation of the vortex string into a helix with curvature 3 μm$^{-1}$ and torsion 11 μm$^{-1}$. The geometric chirality of the vortex string is determined by the magnetic helicity of the vortex texture, constituting coupling of two chiral parameters within the same texture. Beyond the vortex state, we anticipate that complex curvilinear objects hosting 3D magnetic textures like curved skyrmion tubes and hopfions can be characterized by multiple coupled magnetochiral parameters, that influence their statics and field- or current-driven dynamics for spin-orbitronics and magnonics.

Symmetry effects are fundamental in condensed matter physics as they define not only interactions but also resulting responses for the intrinsic order parameter depending on its transformation properties with respect to the operations of space inversion and time reversal. In magnetism, the magnetization vector **M** remains unaltered upon the space inversion symmetry transformation but changes its sign with time inversion. Much attention is devoted to magnetic materials or layer stacks with structural space inversion symmetry breaking, which leads to the appearance of chiral exchange interaction known as the Dzyaloshinskii−Moriya interaction (DMI)[1–5]. The latter manifests itself in the formation of nontrivial chiral and topological spin textures, such as magnetic skyrmions[6–11], bubbles[12], homochiral spin spirals[13], and domain walls[14,15]. Magnetochirality is typically tailored at the intrinsic structural level by the proper selection of specific materials and adjustment of their composition.

Alternatively, space inversion symmetry breaking of the magnetic order parameter appears in geometrically curved systems[16]. In curvilinear ferromagnets, curvature governs the appearance of geometry-induced chiral and anisotropic responses[17–21]. Much attention is dedicated to the exchange interaction, which enables curvature-induced extrinsic DMI as was proposed theoretically[22,23] and validated experimentally[24] for the case of conventional achiral magnetic materials. Thus, geometric curvature of thin films and nanowires is envisioned as a new toolbox to create artificial chiral nanostructures from achiral magnetic materials suitable for the stabilization of skyrmions[25], chiral domain walls[26] and their utilization for prospective spintronic devices[27].

[1]Helmholtz-Zentrum Dresden-Rossendorf e.V., Institute of Ion Beam Physics and Materials Research, Bautzner Landstr. 400, 01328 Dresden, Germany. [2]Institute for Solid State Research, IFW Dresden, 01069 Dresden, Germany. [3]Kyiv Academic University, 03142 Kyiv, Ukraine. [4]Taras Shevchenko National University of Kyiv, 01601 Kyiv, Ukraine. [5]Institute of Solid State and Materials Physics, TU Dresden, 01069 Dresden, Germany. [6]Würzburg-Dresden Cluster of Excellence ct.qmat, Dresden, Germany. ✉e-mail: o.volkov@hzdr.de; d.wolf@ifw-dresden.de; d.makarov@hzdr.de

In addition to the local exchange interaction, the impact of non-local magnetostatic interaction on the properties of curvilinear ferromagnets enables the stabilization of topological magnetic field nanotextures[28], the realization of high-speed magnetic racetracks[29] and curvature-induced asymmetric spin wave dispersions in nanotubes[30]. Furthermore, symmetry analysis demonstrates the possibility to generate a fundamentally new chiral symmetry-breaking effect, which is essentially nonlocal[31]. For the manifestation of the nonlocal chiral symmetry break, there are strict requirements that are imposed both on geometric symmetries and on the magnetic texture. In particular, on the geometry side, the top and bottom surfaces of the object should not be equivalent. In addition, the magnetic texture should have both in- and out-of-plane magnetization components of different parity with respect to the coordinate reflection procedure[31]. Not only the experimental validation of the predicted nonlocal chiral symmetry break is pending but also its consequences for micromagnetic textures are still to be understood. Yet it is expected that the impact should be different compared to the familiar *local* chiral effects induced by the DMI.

The paradigmatic example of a nonlocal texture, which satisfies the above requirements, is the magnetic vortex in a nanodisk[32-35], see Fig. 1a. This spatial distribution of magnetization is characterized by a closed-flux in-plane component with either counter-clockwise or clockwise magnetization rotation, which is referred to as *circulation*, $C = \pm 1$, respectively. Furthermore, the vortex core has a localized out-of-plane magnetization component aligned upward or downward ($P = \pm 1$), determining the vortex *polarity* (see Fig. 1a). The vortex state is topologically nontrivial and characterized locally by the topological charge flux density, $\mathbf{\Omega}$, (see "Methods" and Supplementary Note 1 for details) and by the topological charge (skyrmion number, Pontryagin index), $Q = qP/2$, determined by the product of the polarity, $P$, and vorticity, $q$ ($q = 1$ for a vortex and $q = -1$ for an antivortex)[36,37]. We characterize the global chiral property of the vortex texture via magnetic helicity[38,39]. Being normalized to its absolute value, it reads $\tilde{C} = CP$ (see Supplementary Note 2). The Bloch line in the vortex core we refer as the vortex string, which is a locus of maximal $\Omega = |\mathbf{\Omega}|$. The vortex state in a symmetric disk with equivalent top and bottom surfaces being doubly degenerate with respect to $\tilde{C}$ is situated in the geometric center with the vortex string being a straight line (Fig. 1b;

see also Supplementary Note 3A and Supplementary Fig. 2b). An intrinsic DMI results in a modification of the size of the vortex core and selects an energetically favorable state according to the type and sign of the DMI constant as well as the sign of the vortex circulation $C$[40], which we generalize in the framework of $\tilde{C}$ (Fig. 1c; see also Supplementary Note 3B and Supplementary Fig. 2c). Independent of the presence of the DMI, the locus of the vortex string in planar disks is a straight line. Therefore, similar to local textures in thin films like skyrmions, vortices are characterized by only one magnetochiral parameter. In addition to planar disks, vortices can be spontaneously forming ground states in confined curvilinear shells[41,42], including magnetic caps on spherical nonmagnetic particles[43-45].

Here, we demonstrate experimentally and theoretically the existence of the nonlocal chiral symmetry-breaking effect by studying the deformation of a vortex string in a cap-like asymmetric permalloy (Ni$_{81}$Fe$_{19}$) structure (Fig. 1d; see Supplementary Notes 3B and C, and Supplementary Fig. 3 for details). By combining experimental data and theoretical results, we show that the vortex string can be described as a space curve possessing nonzero curvature, $\kappa_v$, and torsion, $\tau_v$, which is positive (right-handed rotation) for $\tilde{C} = +1$ and negative (left-handed rotation) for $\tilde{C} = -1$ (Fig. 1d). In this respect, vortex texture in a curvilinear object is characterized by multiple magnetochiral parameters. Namely, the chirality of the vortex string and magnetic helicity of the texture. These two magnetochiral properties are linked as the sign of the torsion of the vortex string is determined by the direction of the circulation of the in-plane magnetization component of the vortex for the given vortex polarity. The equilibrium magnetic states of the asymmetric cap are visualized by transmission electron microscopy (TEM) based electron holography revealing the presence of vortices with a positive chirality, $\tilde{C} = +1$, only. The combined theoretical and experimental study allows to identify a new chirality coupling effect where the geometric chirality of the vortex string is determined by the magnetic helicity of the vortex texture.

## Results

### Nonlocal chiral symmetry break for vortex textures

The magnetostatic interaction is responsible for the formation of closed-flux magnetic vortex textures in soft ferromagnetic systems. It is

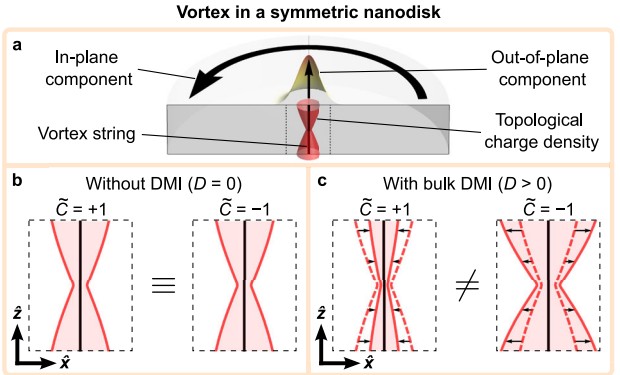

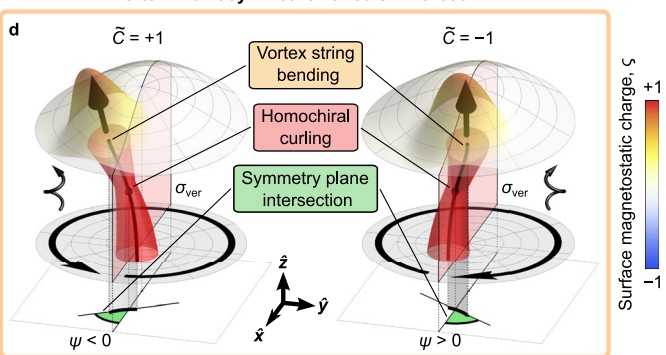

**Fig. 1 | Schematic representation of the influence of symmetry-breaking effects on magnetic vortices.** Magnetic vortex appears in a spatially confined magnetically soft ferromagnetic nanoobject due to the formation of a closed-flux magnetization distribution[33-35]. **a** Schematics of the magnetic vortex in a disk with circulating in-plane and localized out-of-plane magnetization components. The maxima of the distribution of the topological charge density (isosurface is schematically shown by red) determines the location of the vortex string, which is a straight line for the case of a disk with equivalent top and bottom surfaces. **b** Without DMI, the magnetic vortex state is degenerate with respect to the sign of the vortex chirality $\tilde{C} = CP$, which is determined by the product of the circulation $C$ and polarity $P$. **c** DMI lifts the degeneracy of the vortex state with respect to the vortex chirality $\tilde{C}$. For the case of the bulk-type DMI with a positive DMI constant,

the vortex state characterized by $\tilde{C} < 0$ is energetically preferred compared to the higher-energy but stable state with $\tilde{C} > 0$[40]. The vortex core is broader (narrower) for vortices with $\tilde{C}$ of negative (positive) sign compared to the vortex state in achiral disks. Independent of the sign of the DMI constant, the vortex string is straight along the disk axis. **d** We report that absence of the axial and mirror symmetries in the geometry of a magnetic object leads to the deformation of the vortex string due to nonlocal chiral symmetry breaking. Specific features of the vortex string are its bending and inclination by the angle $\psi$ to the mirror symmetry plane. Furthermore, vortex string experiences homochiral curling deformations determined by the vortex chirality $\tilde{C}$, which constitutes coupling of two magnetochiralities within the same texture. The curling deformation of the vortex string can be described as a space curve with nonzero curvature ($\kappa_v$) and torsion ($\tau_v$).

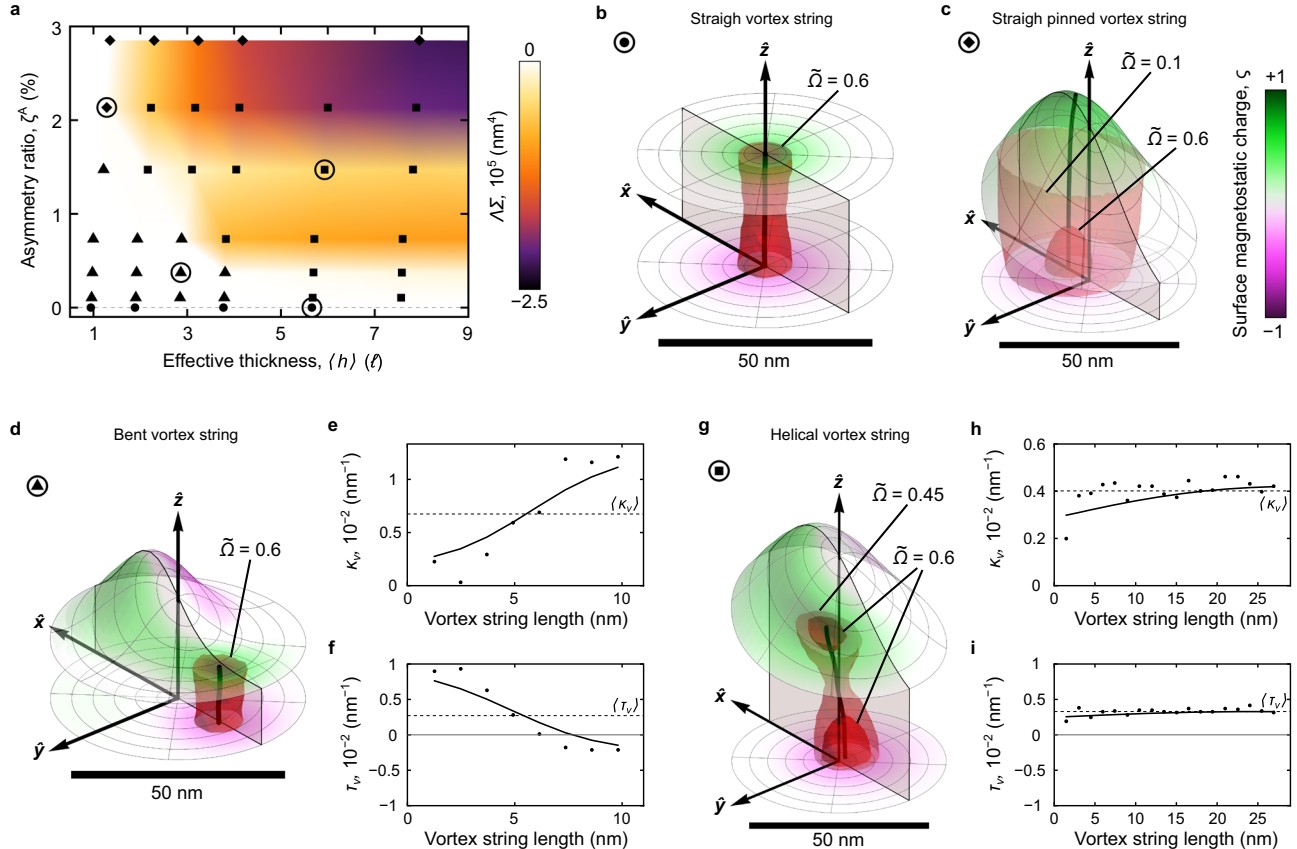

**Fig. 2 | Vortex states in nanodisks of different geometry. a** Product of the total surface and volume magnetostatic charges $\Sigma$ and $\Lambda$, respectively, for different nanodisk geometries. Symbols correspond to the results of full-scale micromagnetic simulations of nanodisks with a radius of 150 nm. States, which are typical for the regions marked by different symbols, are shown in the following panels. **b** The central part of the symmetric nanodisk (thickness: $h = 30$ nm) contains the straight vortex line. Here and below, the red region depicts the isosurfaces for $\tilde{\Omega}$, that determines the spatial localization and shape of the vortex string and illustrate its width profile over the sample thickness. **c** The central part of the highly asymmetric nanodisk of small thickness ($h = 5$ nm), accommodating a tall off-centered Gaussian bump ($t = 50$ nm, $b = 20$ nm and $x_0 = 10$ nm) at its top surface, contains a pinned vortex string with a very small curvature. Here, $\langle \kappa_v \rangle = 0.7$ μm$^{-1}$, which results in a substantial torsion $\langle \tau_v \rangle = 3.7$ μm$^{-1}$ (helix with radius $\langle R_v \rangle = 9.1\ell$ and pitch

$\langle P_v \rangle = 308.2\ell$). **d** The central part of an asymmetric nanodisk ($h = 15$ nm) with a shallower Gaussian bump ($t = 20$ nm, width $b = 10$ nm and shift $x_0 = 10$ nm) on the top surface of the disk contains a short vortex string, which is expelled from the disk center. Distribution of **e** curvature $\kappa_v$ and **f** torsion $\tau_v$ along the vortex string shown in (**d**). Symbols indicate on-site values, solid lines represent trends and dashed lines show mean values. Here, the average string curvature $\langle \kappa_v \rangle = 6.7$ μm$^{-1}$ and torsion $\langle \tau_v \rangle = 3$ μm$^{-1}$. **g** The central part of a thick asymmetric nanodisk with a Gaussian bump ($h = 30$ nm, $t = 40$ nm, $b = 20$ nm and $x_0 = 10$ nm) containing a curled vortex string (c.f. Fig. 1b). Distribution of **h** curvature $\kappa_v$ and **i** torsion $\tau_v$ along the vortex string shown in (**g**). Symbols indicate on-site values, solid lines represent trends and dashed lines show mean values. Here, the average curvature $\langle \kappa_v \rangle = 4$ μm$^{-1}$ and torsion $\langle \tau_v \rangle = 3.3$ μm$^{-1}$.

convenient to analyze the influence of magnetostatics on magnetic states relying on the concept of surface and volume magnetostatic charges $\varsigma$ and $\lambda$, respectively[46]. This allows to decompose the magnetostatic energy in terms of their interaction, namely $\mathcal{E}_{ms} = \mathcal{E}_{\varsigma\varsigma} + \mathcal{E}_{\lambda\lambda} + \mathcal{E}_{\lambda\varsigma}$. The last cross-term, $\mathcal{E}_{\lambda\varsigma}$, binds features of the magnetic distribution on the surface of the sample with magnetization inhomogeneities in the volume, which can induce chiral symmetry break[31]. Being usually zero in the static case of rectilinear geometries, $\mathcal{E}_{\lambda\varsigma} \equiv 0$, this term is highly sensitive to the appearance of any asymmetry in top and bottom surfaces of the magnetic samples. Geometrically curved magnetic thin films offer a straightforward approach to lift the equivalence of the top and bottom interfaces and allow to explore the influence of the $\mathcal{E}_{\lambda\varsigma}$ cross-term on magnetic textures.

To guide the experimental realization, we perform full-scale finite-element micromagnetic simulations in the time domain and compute phase diagrams of the total surface and volume magnetostatic charges $\Sigma$ (Supplementary Fig. 5a) and $\Lambda$ (Supplementary Fig. 5b), respectively, for different symmetric and asymmetric nanodisk geometries. To characterize the geometry, we analyze the difference between the top and bottom surface areas, $S^T$ and $S^B$,

respectively, and the effective thickness of the object $\langle h \rangle = 1/S^B \int_{S^B} dS\, z(\mathbf{r})$ with the integration over the bottom surface and $\mathbf{r}$ being the space coordinate inside the magnetic body. Without loss of generality, we discuss a model system of a soft ferromagnetic disk-shaped object (radius $R$) with a flat bottom surface and the top surface accommodating a Gaussian bump (see "Micromagnetic simulations" under Methods section and Supplementary Note 1 for details). The difference in the surface areas can be tailored by changing the amplitude and width of the Gaussian bump and quantified by the surface asymmetry ratio, $\zeta^A = (S^T - S^B)/(S^T + S^B)$.

In the case of an asymmetric nanodisk with an off-centered Gaussian bump, the total surface and volume magnetostatic charges are both nonzero in a broad range of geometric parameters. Furthermore, their product is negative, which forces the micromagnetic system to distort the vortex structure at equilibrium compared to the flat case (colored region in Fig. 2a; see Supplementary Note 4 and Supplementary Figs. 4 and 5 for details). Unlike a flat symmetric disk with a vortex in the center and $\Lambda\Sigma = 0$ (Fig. 2b), a geometric asymmetry leads to the appearance of $\Lambda\Sigma \neq 0$. As a consequence, the interaction between the surface and volume magnetostatic charges acts on the vortex string by

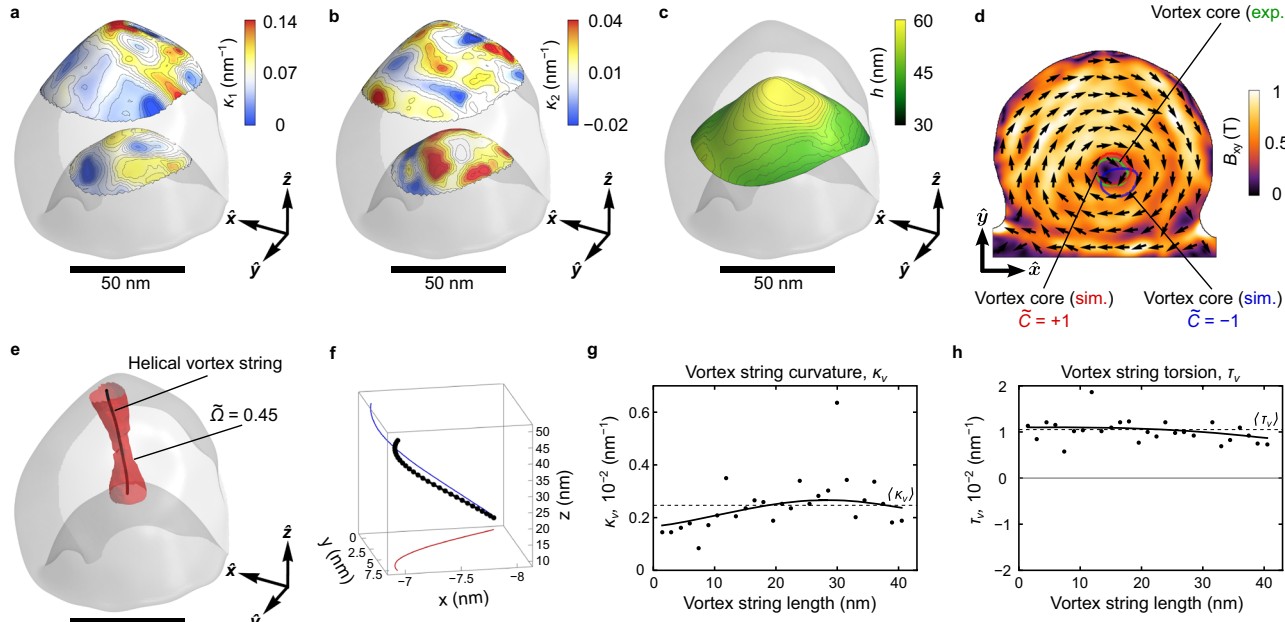

**Fig. 3 | Experimental and micromagnetic investigation of a permalloy nanocap.** Distribution of the principal curvatures **a** $\kappa_1$ and **b** $\kappa_2$ on the top and bottom surfaces of the tomographically reconstructed nanocap structure. The color scheme represents the change of the curvature value. **c** Thickness distribution on the equidistant surface inside the central part of the cap. **d** Map of the magnitude of the magnetic induction in the $\hat{x}\hat{y}$ projection reconstructed by the off-axis electron holography measurements. The color scheme and black arrows indicate the modulus and direction of the in-plane **B**-field component, respectively. The green area shows the position of the projection of the vortex string derived from experimental data: the red and blue areas indicate the $\hat{x}\hat{y}$ projections of vortex strings obtained in full-scale micromagnetic simulations for $\tilde{C} = +1$ and $\tilde{C} = -1$, respectively. **e** The spatial profile of the vortex string with $\tilde{C} = +1$ and the distribution of the density of the topological charge (isosurface of $\tilde{\Omega} = 0.35$). **f** Scaled form of the vortex string extracted from the center of mass of the topological charge $\tilde{\Omega}$, see Supplementary Note 1 for more details. **g** Curvature and **h** torsion along the vortex string shown in (**e**). Symbols and solid lines correspond to the on-site values and trend, respectively. Dashed line shows mean values $\langle \kappa_v \rangle$ and $\langle \tau_v \rangle$.

pinning it at corrugation (Fig. 2c), pushing it out from the geometrical center (Fig. 2d–f), or deforming it with a homochiral twist (Fig. 2g–i). Thus, the magnitude of the product of the total surface and volume charges $\Lambda\Sigma$ can be used as a quantitative measure of the strength of the nonlocal chiral interaction. Namely, in the case of large surface asymmetry ($\zeta \gtrsim 2\%$), the vortex string is located in the center of the Gaussian bump and remains almost straight with the curvature $\langle \kappa_v \rangle = 0.7\ \mu m^{-1}$ and torsion $\langle \tau_v \rangle = 3.7\ \mu m^{-1}$ (the straight line is a limiting case of a helix with $\kappa \to 0$ and $\tau \to \infty$, that corresponds to a helix with radius $R_v \to 0$ and pitch $P_v \to \infty$, see Fig. 2c and Supplementary Fig. 6). This behavior is driven by the dominant contribution of surface magnetostatic charges, which produce an effective easy axis out-of-plane anisotropy along $\hat{z}$ of the bump that was studied in detail in ref. [47] for the case of asymmetric yet ultrathin magnetic shells with a strong roughness.

In the opposite case of a thin sample ($\langle h \rangle \approx 3\ell$ with $\ell = 5.3$ nm being the exchange length for permalloy) with a small surface asymmetry ($\zeta \approx 0.5\%$), the vortex core is displaced away from the disk center opposite to the bump. The resulting vortex string experiences a bending deformation in the direction opposite to the center of the Gaussian bump and is characterized by the curvature $\langle \kappa_v \rangle = 6.7\ \mu m^{-1}$ and the torsion $\langle \tau_v \rangle = 3\ \mu m^{-1}$, which corresponds to a helix with radius $R_v = 24.1\ell$ and pitch $P_v = 60.4\ell$ (see Fig. 2d–f and Supplementary Fig. 7). This deformation of the vortex string appears together with $\Lambda\Sigma \gtrsim 0$, due to the optimization of the magnetostatic energy with respect to the imbalanced surface magnetostatic charges on the top and bottom surfaces. Moreover, the loss in volume charges is compensated by a strong energy gain in exchange due to the short length of the vortex string. The displacement of the texture is driven by magnetostatic effects, which is distinct from the previously considered displacement of domain walls[48] and skyrmions[49] in curvilinear geometries due to the local chiral interactions. We note that the local theory of the curvilinear

magnetism[17,25] is valid within a narrow region of the phase diagram Fig. 2a along the $x$ axis below the effective thickness of about $3\ell$. It should be noted that bending of the vortex string is not accompanied by its branching. This is in contrast to the observed evolution of skyrmion tubes into a bifurcated complex 3D texture[50–52].

For other asymmetric geometries on the phase diagram with $\zeta \gtrsim 1\%$ and $\langle h \rangle \gtrsim 3\ell$, the equilibrium vortex string is significantly deformed (Fig. 1g). This region of geometric parameters corresponds to moderate values of the product of $\Lambda$ and $\Sigma$, see Fig. 2a, and it is characterized by the optimization of the magnetostatic energy through the vortex string deformation inside the magnetic body. The latter can be described as a space curve with pronounced curvature, $\kappa_v$, and torsion, $\tau_v$, see Figs. 1h and 2i, respectively. Furthermore, for the specific sign of the vortex chirality $\tilde{C}$, there is only one stable configuration: the vortex string can be described either as a curve with $\tau_v > 0$ (right-handed rotation) for $\tilde{C} = +1$ or $\tau_v < 0$ (left-handed rotation) for $\tilde{C} = -1$. The helicity-dependent twist of the vortex string is a consequence of the nonlocal chiral symmetry break. We note that the geometric deformation of the vortex string in a helix constitutes the appearance of a second magnetochiral parameter for the vortex texture. Furthermore, the two magnetochiral parameters appear to be coupled. This region of the phase diagram corresponding to the moderate product of $\Lambda$ and $\Sigma$ is chosen for the experimental validation of the predicted effects.

**Experimental validation of nonlocal chiral symmetry break**

To experimentally realize a nanostructure with the required asymmetry ratio and effective average thickness exhibiting sizeable nonlocal chiral effects (Fig. 1a, g), we deposited a 50-nm-thick permalloy film on nonmagnetic polystyrene spheres with a diameter of 80 nm. Bright-field TEM tomography[53,54] was applied to reveal the 3D shape of the cap structure, whose distribution of principal curvatures and

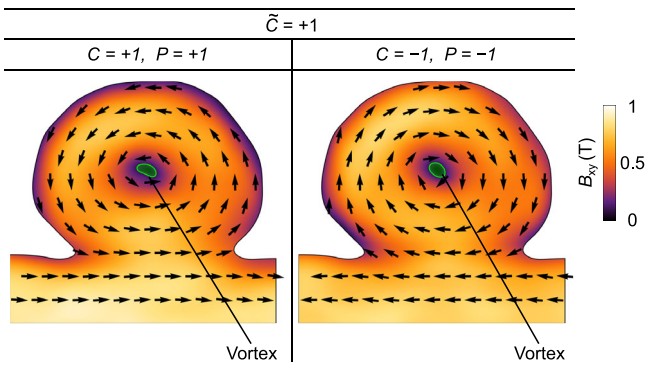

**Fig. 4 | Experimental equilibrium magnetic vortex states in permalloy caps.** The figure shows maps of the magnetic induction (magnitude and direction of the $\boldsymbol{B}$-field) in the $\hat{\mathbf{x}}\hat{\mathbf{y}}$ projection obtained by means of off-axis electron holography for magnetic vortices with $C = +1, P = +1$ and $C = -1, P = -1$. We stress that only vortices with $\tilde{C} = +1$ are observed experimentally.

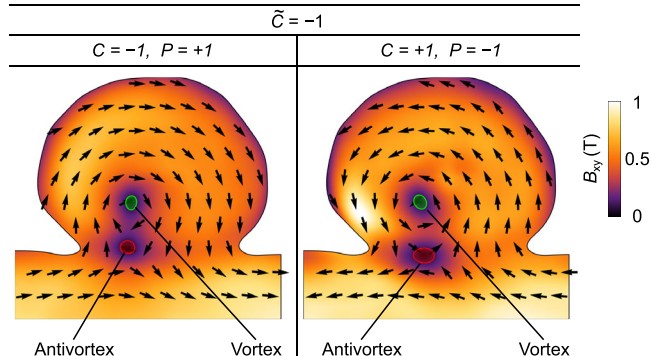

**Fig. 5 | Experimental topologically trivial ($Q = 0$) magnetic states with vortex–antivortex pairs in asymmetric caps.** The figure shows maps of the magnetic induction (magnitude and direction of the $\boldsymbol{B}$-field) as in Fig. 4. The change of the vortex chirality and polarity causes the formation of topologically trivial states, where the topological charge of magnetic vortex ($Q = \pm 1/2$) is compensated by the charge of antivortex ($Q = \mp 1/2$).

thickness are shown in Fig. 3a–c, respectively (see Supplementary Note 6 for more details). The resulting cap has nonequivalent top and bottom surfaces, which is quantified by the two different principal curvatures (Fig. 3a, b). The respective asymmetry ratio between the area of the top and bottom surfaces is about 1.8%, while the effective thickness of the film, $\langle h \rangle$, is about $5.0\ell$. For these geometric parameters, the experimental sample is well placed within the range where the vortex string is expected to show a sizeable curling deformation, see Fig. 2.

Magnetic imaging is done by means of off-axis electron holography[55], which reveals the projections of the in-plane magnetic induction distribution, $B_{xy}$, on the $\hat{\mathbf{x}}\hat{\mathbf{y}}$ plane, see Fig. 3d. The data shows the presence of the off-centered vortex state with in-plane circulation $C = -1$, which denotes the clockwise magnetization rotation in the $\hat{\mathbf{x}}\hat{\mathbf{y}}$ plane. The state is prepared in a way to assure having a vortex with negative polarity ($P = -1$), resulting in a positive vortex chirality, $\tilde{C} = +1$. The vortex string is localized at the position, where the cap is the thickest (compare Fig. 3c, d). Furthermore, we observe that the shape of the projected vortex string is not circular and reveals sizeable ellipticity ($\epsilon = p_2/p_1 = 0.86$ for $p_1$ and $p_2$ being semi-major and semi-minor axes, respectively). This can be either due to the deformation of the in-plane vortex texture, or due to the projection of the bent vortex string to the $\hat{\mathbf{x}}\hat{\mathbf{y}}$ plane. To understand the details of the contrast, we perform finite-element micromagnetic simulations of vortex states with different vortex chiralities (Supplementary Figs. 11–14). For these simulations, the object geometry was retrieved from the tomographic reconstruction (see Fig. 3a–c and Supplementary Note 6 for the details). The simulations confirm the second assumption that the vortex string is geometrically deformed (see the shape of the vortex string of positive chirality $\tilde{C}$ in Fig. 3e, f and the comparison of projections of vortex strings of different chiralities with the experimental result in Fig. 3d). The curvature of the vortex string increases from bottom surface of the nanocap to the top one with $\langle \kappa_v \rangle = 2.5 \ \mu m^{-1}$, while the torsion remains constant along the string with $\langle \tau_v \rangle \sim 11 \ \mu m^{-1}$, see Fig. 3g, h, Supplementary Note 6C, and Supplementary Table 2. The positive sign of torsion for the positive magnetochirality $\tilde{C}$ of the vortex is in line with the predicted chirality coupling effect, which links two magnetochiral parameters of the texture, c.f. Fig. 2g–i.

To understand the formation of magnetochiral vortex states, we performed additional experimental investigation of another asymmetric nanocap possessing vortices of different circulation and polarity. Herein, the polarity is manipulated by deliberately varying the external magnetic field with the help of the objective lens of the TEM. Note, however, that in all switching trials, we only observe vortices of $\tilde{C} = +1$, which correspond to $P = +1, C = +1$ and $P = -1, C = -1$ (Fig. 4). Consequently, the circulation of the vortex texture is linked to the

vortex polarity in a specific way always yielding a positive magnetic chirality. Vortex states corresponding to the opposite chirality $\tilde{C} = -1$ ($P = +1, C = -1$ and $P = -1, C = +1$) are not identified in the experiment. Instead, we observed vortex–antivortex pairs restoring a topologically trivial state with $Q = 0$ consisting of a vortex with $Q = \pm 1/2$ and an antivortex with $Q = \mp 1/2$, see Fig. 5. We note that the spatial localization of the vortex–antivortex pairs as well as the distance between these textures are different for different polarities and circulations of the vortex state (but for the same $\tilde{C} = -1$). This difference and its relation to the nonlocal chiral effect is an appealing independent study by itself.

## Discussion

The physics of the geometric deformation of the vortex string related to the vortex chirality can be analyzed in terms of geometric symmetries and balance between the contribution of surface and volume magnetostatic charges. The high-symmetry state is supported by a planar circular nanodisk (Supplementary Note 3A). Taking into account time inversion operation, vortices with the same circulations and opposite polarities are enantiomorphs (mirror images), see Fig. 6a. In this case, the cross-term $\mathcal{E}_{\lambda_\varsigma} = 0$, as it depends on both spatial derivatives and surface normals. The presence of DMI lifts the energetic degeneracy between the states with different $\tilde{C}$ but does not change the vortex string configuration (Supplementary Note 3B).

By changing the shape of the top surface in the form of a centered Gaussian bump, we find from simulations that the vortex core is not located at the geometric center of the disk (Supplementary Note 3C). The vortex string is bent along the thickness of the disk. The top and bottom equilibrium positions of the vortex string are off-centered with different radii with respect to the geometrical center (Fig. 6b). This means that there are infinite angular positions around the geometric center for which the magnetic state is degenerates. Therefore, the system supports an infinite manifold of equilibrium vortex states (Supplementary Fig. 3a–d). This is equivalent to the appearance of a zero-frequency Goldstone mode for a vortex string excited along a circular trajectory around the disk center. A further analysis of the locus for the vortex string along the thickness of the disk reveals that besides bending of the vortex string, the curling deformation of the vortex string is also present. This deformation is magnetochiral-dependent: the curling direction is determined by the vortex chirality being right-handed (i.e., $\tau_v > 0$) for $\tilde{C} = +1$ and left-handed (i.e., $\tau_v < 0$) for $\tilde{C} = -1$. In other words, for a given polarity and in-plane circulation, only one of the curling directions of the vortex string exists.

In the case of a nanodisk without the axial symmetry but with the vertical mirror symmetry plane $\sigma_{ver}$ (Fig. 6c; i.e., the Gaussian bump is

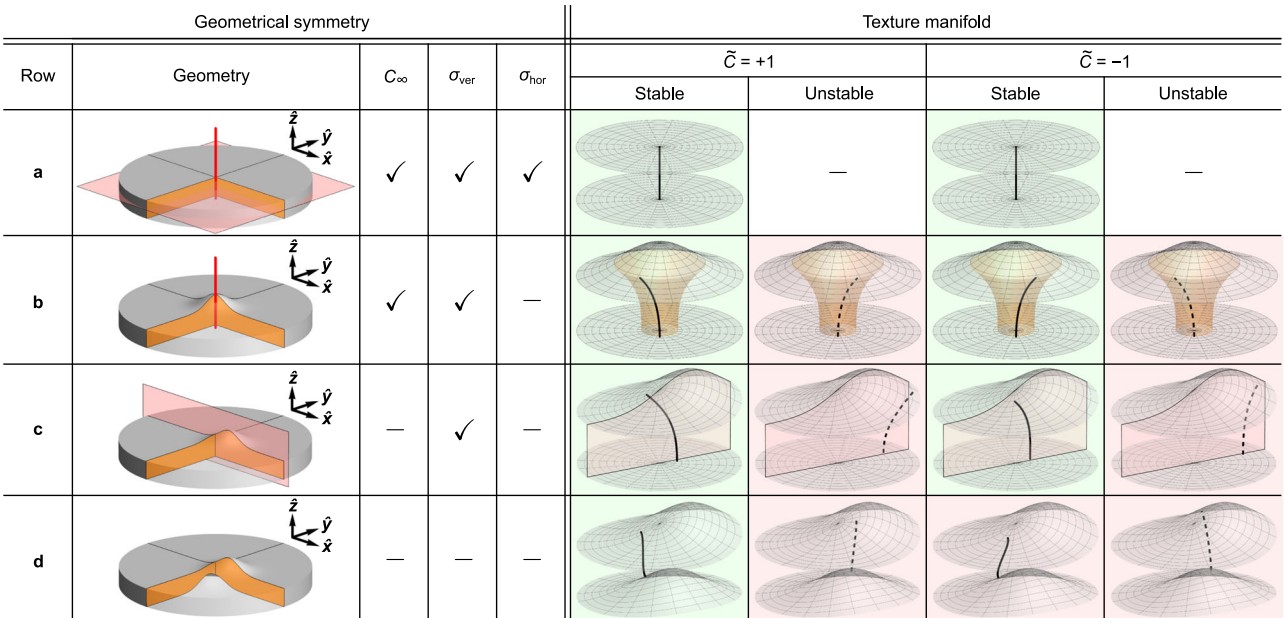

**Fig. 6 | Schematics of the evolution of the shape of the vortex string in magnetic nanodisks with a Gaussian bump. a** Planar magnetic disk possesses axial $C_\infty$ symmetry, one horizontal $\sigma_{hor}$ and infinite number of vertical mirror planes $\sigma_{ver}$. Such system supports two equilibrium vortices with $\widetilde{C} = \pm 1$, that possess the same straight vortex strings without any deformation. These states can be obtained by $\sigma_{hor}$ with taking into account the time inversion operation. **b** Removing the $\sigma_{hor}$ mirror plane, the vortex core at the top surface displaces thereby introducing bending deformation with additional helicity-dependent curling of the vortex string. The sense of curling is specific for the particular vortex chirality. Such deformation of the vortex string can take any angular position at remanence, due to

the axial symmetry in the system. Dotted lines correspond to unstable vortex string deformations unfavorable for the particular chirality $\widetilde{C}$. The green- and red-filled cells correspond to the energetically favorable and unfavorable states, respectively. **c** If the $C_\infty$ symmetry is also absent (for instance, due to the presence of a shifted Gaussian bump), only two stable vortices can be formed: one vortex with $\widetilde{C} = +1$ and right-handed curling of the vortex string, and another vortex with $\widetilde{C} = -1$ and left-handed curling. **d** In the case of a fully asymmetric nanodisk, there is only one equilibrium vortex state with $\widetilde{C} = +1$. This is due to the metastability of vortices with chirality $\widetilde{C} = -1$ as they provide longer vortex strings.

shifted out of the center of the top surface), only one stable angular position of the vortex string exists for a given vortex chirality. Thus, vortices with $\widetilde{C} = +1$ and $\widetilde{C} = -1$ have the same energy, but are differently localized in space. Still, the vortex string remains deformed with the same curvature but different signs of the torsion, see Figs. 1c and 6c (see also Supplementary Fig. 3e–h). As a result of the symmetry lowering and the well-defined non-degenerate equilibrium state, the Goldstone mode vanishes.

Mimicking the experimental situation with a cap of low symmetry, we finally consider two off-centered Gaussian bumps at the top and bottom surfaces of the nanodisk. This shape eliminates the $\sigma_{ver}$ symmetry of the previous case (Fig. 6d). As a consequence, the system supports only one equilibrium vortex state ($\widetilde{C} = +1$ in Fig. 6d) that corresponds to the shortest vortex string. In line with the theoretical predictions, we observe experimentally that vortices with the same positive chirality, namely different sign for the polarity and circulation, have the same spatial positions and homochiral deformations of the vortex string (Fig. 4).

To understand the role of the exchange and magnetostatic interactions on the formation of curved vortex strings, we perform additional micromagnetic simulations where magnetostatic interaction was switched off and/or replaced with effective anisotropy models, see details in Supplementary Note 5 and Supplementary Table 1. In the first model case, we utilize only exchange interaction and the magnetization is pinned at the surface facet of the mesh to assure the vortex state, while the magnetostatic interaction is switched off. Thus, the magnetic system with only exchange interaction minimizes the vortex string length going outside the bump to the region with flat top and bottom surfaces. As this observation is not in line with the experiment, we conclude that accounting for surface magnetostatic charges at the surface facet only is insufficient. The

resulting vortex string has $\langle \kappa_v \rangle = 2.0\,\mu m^{-1}$, which is three times smaller than the one obtained for the full-scale simulation $\langle \kappa_v \rangle = 6.2\,\mu m^{-1}$, while the vortex string torsion $\langle \tau_v \rangle = -6.6\,\mu m^{-1}$ has opposite sign to the case of the full-scale simulation $\langle \tau_v \rangle = 9.0\,\mu m^{-1}$. Furthermore, the vortex core dimensions increase in the absence of magnetostatic interaction.

In the second model case, we consider magnetization to be pinned in the vortex state at all sample boundaries in a system with exchange interaction and additional shape anisotropy, that replace magnetostatics. We perform simulations for two model cases of anisotropy distributions: (i) homogeneous easy-plane anisotropy in the $\hat{x}\hat{y}$ plane and (ii) spatially varying "in-surface" anisotropy, with the easy surface of magnetization linearly changing from the bottom to the top surfaces. Both these anisotropy models reveal that the equilibrium vortex position lies outside the bump with a substantial shrinking of the vortex core radius to a radius of about mesh size. Still, both models provide similar curvature being $\langle \kappa_v^{(i)} \rangle = 3.2\,\mu m^{-1}$ and $\langle \kappa_v^{(ii)} \rangle = 3.3\,\mu m^{-1}$, while the torsion values are different being $\langle \tau^{(i)} \rangle = 4.3\,\mu m^{-1}$ and $\langle \tau^{(ii)} \rangle = 1.3\,\mu m^{-1}$. These inconsistencies with the experimental observations indicate that the shape anisotropy coming from surface magnetostatic charges itself is insufficient to describe the position of the vortex string and its shape within the Gaussian bump. The role of the volume magnetostatic charges is decisive. Hence, the homochiral deformation of the vortex string as well as the state selection due to the geometrical symmetry reductions originates from the dipole-dipole interaction and the related nonlocal chiral symmetry break. Accordingly, the experimental results validate the existence of the nonlocal symmetry-breaking effect predicted by the generalized micromagnetic theory of curvilinear shells[31].

The discussed nonlocal chiral effects are distinct from the *exchange-driven polarity-circulation coupling* and *boundary effects*

induced by the sample shape. The first type of effects originates in thin shell systems, where curvature of the surface introduces a coupling between the localized out-of-surface part of the magnetization texture and its delocalized in-surface component as was predicted for vortices in spherical shells[56] and merons in parabolic and hyperbolic shells[57]. The boundary effects are expected in magnetic systems without axial symmetry, which introduces stable positions for multiple equilibrium magnetic distribution. Namely, elliptical-[58] or ellipsoid-shape[59,60] particles exhibit a deformed vortex core, which connects two geometric focal points through the formation of a Bloch domain wall with the Néel cap[37]. Such magnetization distribution originates from the interplay between the exchange and magnetostatic energy, which tends to optimize magnetization distributions by the formation of divergence-free solenoidal textures. Thus, the resulting textures are bi-stable as a vortex core connects pairwise any of the two focal points on different surfaces of the sample, as these states are symmetric and energetically equivalent.

The consequence of the nonlocal chiral symmetry break is the new chirality coupling effect. When we discuss on the geometric chirality coupling, geometry-governed chiral interactions can realize an interplay between the geometric chirality of the object and the magnetochirality of the texture with nontrivial magnetic helicity[21]. Typically, this means that the geometric chirality of the sample determines the magnetochirality of the texture. Now we observe another phenomenon: the sign of the torsion of the vortex string (i.e., chirality of its locus still characterizing the magnetic texture) is determined by the sign of the circulation of the vortex state for the given polarity (i.e., magnetic helicity of the vortex). In this respect, the magnetic helicity of the texture determines the geometric chirality of the vortex string. This is different to the vortex in a planar disk, where the vortex string is straight and its shape is independent of the circulation. In the asymmetric sample, both parameters of the texture are chiral and being coupled.

In summary, we demonstrated that the geometric asymmetry is a key ingredient for the observation of the nonlocal symmetry-breaking effect in a magnetic object. This effect originates from the interplay between surface and volume magnetostatic charges. The nonlocal chiral symmetry breaking is detected experimentally through the stabilization of nontrivial magnetic textures of a specific magnetic chirality, which is determined by the sample's symmetry. Although geometrically curved objects offer a convenient playground to observe nonlocal chiral effects, we stress that the conclusions of this work are generic and valid not only for curved three-dimensional architectures. These effects are also expected in any object possessing nonequivalent opposite surfaces, whose size allows the presence of nonvanishing volume magnetostatic charges and hosting a noncollinear magnetic texture. For instance, vortices in wedge-shaped disks could experience a bent of the vortex string as discussed above. Furthermore, thin metal films deposited on flat substrates like Si wafers usually possess corrugated top surfaces. Therefore, it would be insightful to analyze the impact of the nonlocal chiral effects on noncollinear textures resting or propagating in these corrugated thin films, which might be relevant for prospective spintronic and spin–orbitronic devices including domain wall and skyrmion-based racetrack memory[61–63]. In this respect, we note that the discussed nonlocal chiral effects also affect local noncollinear textures like domain walls as schematically shown for a Néel domain wall in Fig. 1h in ref. [31]. Still, detailed studies are needed to quantify how large the effects and the physical consequences for the statics and dynamics of domain walls are.

Deformations of a texture should not necessarily be expected as a homochiral or unidirectional in a general sense. Namely, microwave excitations of chiral skyrmion tubes demonstrate both sense of left- and right-handed rotations independently of the skyrmion chirality[64,65]. This is in contrast to our finding that even at equilibrium and in the absence of the Dzyaloshinskii–Moriya interaction, the vortex string not only bends into helix, but also the helix chirality is unique for the given vortex symmetry. It should be noted, that the similar curling deformation of the same chirality has been recently reported for skyrmion tubes in the asymmetrically shaped helimagnetic samples[66]. Thus, we anticipate that our findings are generic for a wide class of magnetic nanotextures and provide an explanation of their structure in geometrically asymmetric samples.

It is natural to extend the presented approach of the formation and analysis of the textures with multiple magnetochiral parameters on more complex geometries hosting 3D textures. For instance, even for localized chiral textures like skyrmions in confined geometries, the sense of axial modulation of the skyrmion strings could be related to the chirality of skyrmion texture themselves[67]. This coupling between chiral parameters can be even more pronounced in magnetization dynamics, e.g., dynamic helical solitary waves propagating along skyrmion strings[64,65]. Furthermore, 3D textures with nontrivial Hopf index $\widetilde{H}$ could offer another appealing system to search for multiple coupled magnetochiral parameters with examples of double rings with $\widetilde{H} = 1$[68] or trefoil knot-like shapes with $\widetilde{H} = 7$[69].

In addition, we anticipate active prospective research on the realization of complex 3D objects accommodating 3D magnetic textures with multiple and coupled magnetochiral parameters relying on direct nanoscale writing[28]. The characterization of these 3D textures and their evolution under external stimuli will require advanced imaging techniques relying on electron holography[67] or X-ray-based tomography approaches[70]. In this respect, curvilinear and 3D magnetic low-dimensional architectures offer an appealing material science platform to fabricate and study novel physics of 3D magnetic textures characterized by multiple magnetochiral parameters, which can be tailored on demand relying on geometric parameters.

## Methods

### Magnetostatics

As the rigorous account of its self-consistent stray field requires the solution of integrodifferential equations, it is convenient to analyze spatial distributions of surface and volume magnetostatic charges[37], $\varsigma$ and $\lambda$, respectively:

$$\varsigma(\mathbf{r}) = \mathbf{m}(\mathbf{r}) \cdot \mathbf{n}(\mathbf{r}), \quad \lambda(\mathbf{r}) = -\nabla \cdot \mathbf{m}(\mathbf{r}), \tag{1}$$

where $\mathbf{m} = \mathbf{M}/M_s$ is the normalized magnetization vector with $M_s$ being the saturation magnetization and $\mathbf{n}$ being the outer surface normal. The magnetostatic energy is determined by the interaction of $\varsigma$ and $\lambda$. The resulting magnetostatic energy, $E_{\mathrm{ms}}$, originates from the pairwise interactions between $\varsigma$ and $\lambda$, which follows from

$$\underbrace{\frac{E_{\mathrm{ms}}}{M_s^2}}_{\mathcal{E}_{\mathrm{ms}}} = \underbrace{\frac{1}{2} \int_S \int_S \mathrm{d}S \mathrm{d}S' \frac{\varsigma(\mathbf{r})\varsigma(\mathbf{r}')}{|\mathbf{r} - \mathbf{r}'|}}_{\mathcal{E}_{\varsigma\varsigma}} + \underbrace{\frac{1}{2} \int_V \int_V \mathrm{d}\mathbf{r} \mathrm{d}\mathbf{r}' \frac{\lambda(\mathbf{r})\lambda(\mathbf{r}')}{|\mathbf{r} - \mathbf{r}'|}}_{\mathcal{E}_{\lambda\lambda}}$$
$$+ \underbrace{\int_V \int_S \mathrm{d}\mathbf{r} \mathrm{d}S' \frac{\lambda(\mathbf{r})\varsigma(\mathbf{r}')}{|\mathbf{r} - \mathbf{r}'|}}_{\mathcal{E}_{\lambda\varsigma}}. \tag{2}$$

Here, the first term $\mathcal{E}_{\varsigma\varsigma}$ describes the interaction between the surface charges at different surfaces of the object. In the case of thin symmetric shells, this term could be reduced to a local shape anisotropy $\mathcal{E}_{\varsigma\varsigma} = 2\pi h \int_S \mathrm{d}S (\mathbf{m} \cdot \mathbf{n})^2$, which linearly depends on thickness $h$ for curved shells[71] and films[72]. The second term $\mathcal{E}_{\lambda\lambda}$ denotes the interplay between volume magnetostatic charges, while the third one $\mathcal{E}_{\lambda\varsigma}$ originates from the pairwise interaction between volume and surface magnetostatic charges. The cross-interaction $\mathcal{E}_{\lambda\varsigma}$ in Eq. (2) is zero in many standard examples of nanomagnets because of

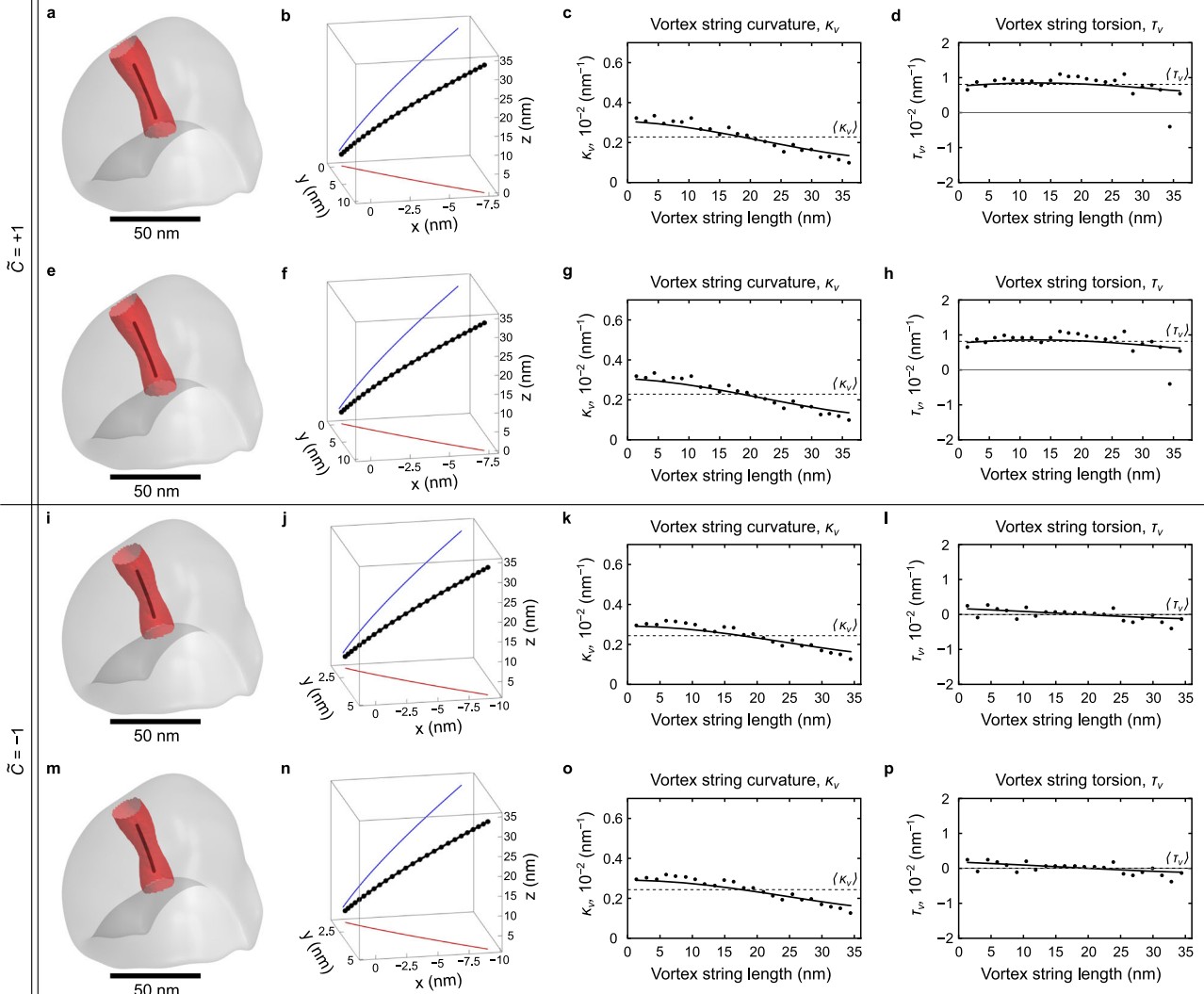

**Fig. 7 | Vortex states in a truncated experimental nanocap.** Panels **a**, **e**, **i**, **m** represent the reconstructed vortex lines (black tubes) inside the truncated asymmetric nanocap with vortices of different magnetic helicity: **a** for $P = +1$, $C = +1$; **e** for $P = -1$, $C = -1$; **i** for $P = +1$, $C = -1$; **m** for $P = -1$, $C = +1$. Their rescaled shapes extracted from the calculation of $\widetilde{\Omega}$ are shown in (**b**, **f**, **j**, **n**), respectively. The resulting distributions of vortex string curvature and torsion are shown in (**c**, **g**, **k**, **o**) and (**d**, **h**, **l**, **p**), respectively. All isosurfaces are constructed for $\widetilde{\Omega} = 0.35$.

two factors: (i) tangential magnetization at the sample edges leads to the absence of respective surface charges; (ii) equivalence of surface charges at the top and bottom magnetic surfaces. The latter is reasonable for conventional soft ferromagnetic planar thin films, where according to the pole avoidance principle, magnetostatics determines the formation of closed-flux magnetic textures[37,46]. In the case of symmetric nanodisk, this leads to the appearance of the vortex distribution with both volume and surface magnetic charges tend to be balanced by opposite ones in equilibrium, which corresponds to $\Sigma = \int_S dS\, \varsigma(\mathbf{r}) = 0$ and $\Lambda = \int_V d\mathbf{r}\, \lambda(\mathbf{r}) = 0$[37]. Thus, although the total surface and volume magnetostatic charges ($\Sigma$ and $\Lambda$, respectively) do not enter to the expression for the magnetostatic energy (2), these quantities are convenient for the analysis of the impact of the integrals in Eq. (2), which contain their local counterparts ($\varsigma$ and $\lambda$, respectively). It should be noted, that the influence of the term $\mathcal{E}_{\lambda\varsigma}$ on localized textures like domain walls is expected in geometries possessing notches, where the requirement $\varsigma_{\text{edge}} = 0$ is not necessarily fulfilled[73]. Furthermore, extended curvilinear magnetic shells naturally break the geometric symmetry between the top and bottom surfaces, resulting in a nonzero $\mathcal{E}_{\lambda\varsigma}$[31].

## Micromagnetic simulations

Full-scale micromagnetic simulations for the experimental geometry and asymmetric nanodisks are performed by means of a finite-element micromagnetic code, the successor of the GPU accelerated TETRAMAG[74,75]. Simulations are done for a magnetic body with micromagnetic parameters of permalloy: saturation magnetization $\mu_0 M_s = 1.08$ T, where $\mu_0$ is the vacuum permeability, exchange constant $A = 13$ pJ/m and exchange length $\ell = \sqrt{2A/(4\pi M_s^2)} = 5.3$ nm. Using the energy minimization approach by means of a conjugate gradient method, the equilibrium magnetic distributions are obtained starting from the initial states corresponding to geometrically-centered vortex states with various polarities, $P = \pm 1$, and circulations, $C = \pm 1$. The calculations are carried out for model nanodisks as well as for experimentally reconstructed permalloy cap structures. As nanodisks, we consider reference objects with flat top and bottom surfaces as well as those with flat bottom surface and the top surface accommodating a Gaussian bump. For nanodisks, we simulate objects with a radius of $R = 150$ nm and thickness in the range $h = [2.5; 60]$ nm. Different structural asymmetries are introduced through the formation of a Gaussian bump at the top surface

of the disk

$$z = t \exp\left[-\frac{(x-x_0)^2 + y^2}{2b^2}\right]. \tag{3}$$

Here, $t = [0; 60]$ nm is the bump height, $b = 20$ nm is its width, and $x_0 = 10$ nm is the bump shift with respect to the center of the disk.

In the case of the experimental geometry, the precise sample shape is obtained by means of bright-field TEM tomography with 0.5-nm spatial resolution in all dimensions. To resemble the experimentally observed vortex state, we introduce fixed boundary conditions for the region corresponding to the magnetic layer on the supporting part. This allows us to reproduce the experimentally obtained homogeneous magnetization distribution, that is formed in the supporting part of the sample (see Supplementary Fig. 10). To investigate the influence of this supporting part on the vortex state and discussed nonlocal chiral effects, we perform additional micromagnetic simulations for the truncated magnetic cap without the supporting part, see Fig. 7, Supplementary Note 6 and Supplementary Table 3. As a result, the magnetic vortex string becomes slightly shifted, but obtains a similar homochiral curling deformation to that observed for the full experimental geometry. This confirms that the main source of the observed nonlocal chiral effects is the asymmetry of the top and bottom surfaces of the object as well as the interplay between the surface and volume magnetostatic charges.

In addition, we utilize MuMax3 code[76] for the simulations of the planar nanodisk (radius $R = 150$ nm and thickness $h = 20$ nm) with DMI to reproduce the vortex chirality splitting obtained in ref. [40]. Such simulations are performed for a permalloy disk with DMI of the bulk type with $D = 0.2$ mJ/m². The equilibrium states are calculated from the initial vortex distributions of various circulation and polarity by means of the conjugate gradient method. Also, we perform additional simulations using MAGPAR code[77] for asymmetric permalloy nanocaps to analyze the influence of the exchange and magnetostatic interactions on the formation of the curved vortex strings, see Supplementary Note 5 for details.

## Determination of the vortex string from topological charge density

A standard approach to determine the vortex string position and its fine structure in thin rectilinear samples is based on the localization of the intersection of isosurfaces $m_x = 0$ and $m_y = 0$[78]. However, this approach meets challenges for the case of thick asymmetric and/or curvilinear magnetic geometries, as magnetic textures may exhibit additional bending that goes beyond the standard Cartesian description. Thus, it is instructive to analyze the distribution of the flux density of the topological charge[68,79,80] over the 3D geometry:

$$\Omega_l = \frac{1}{8\pi} \epsilon_{lno} \epsilon_{ijk} m_i \partial_n m_j \partial_o m_k, \tag{4}$$

where $m_i$ is the normalized local magnetization component, $\epsilon_{ijk}$ is the Levi–Civita tensor and $i,j,k,l,n,o = \{x,y,z\}$. The closed integral over the sample's total surface results in the topological charge (skyrmion number or Pontryagin index) being $Q = \int_S dS \cdot \Omega$, which defines the mapping degree of the magnetization distribution onto a sphere $\mathbf{m}^2 = 1$[79,81,82]. Namely, for the vortex (winding number $q = +1$) and antivortex ($q = -1$) states of positive polarity ($P = +1$) in soft-ferromagnets, the target sphere is half wrapped, which denotes $Q = +1/2$ for the vortex and $Q = -1/2$ for the antivortex, respectively[83].

Spatial derivatives in Eq. (4) lead to the appearance of nonzero topological charge densities $\Omega = |\mathbf{\Omega}|$ in the vicinity of nontrivial magnetization textures, that provide noncollinear magnetization distributions. Thus, extrema in the distribution of $\Omega$ unambiguously determine the position and spatial configuration of the center part of

the topologically nontrivial magnetization textures even in the bent state[70]. As magnetic vortex strings could have different spatial configurations, it is more convenient to introduce the normalized flux density of the topological charge $\widetilde{\Omega} = \Omega/\Omega_{max}$, where the $\Omega_{max}$ is the absolute maximum value of $\Omega$ for the particular magnetization distribution.

It should be emphasized that the introduced in the Eq. (4) the topological charge flux density obtains the same form of the mapping Jacobian[84], previously introduced for the normalized 3D vector field defined on a 3D closed surface[25]. In the limit case of 2D plane systems, it transforms into a gyrocoupling vector (topological density, topological current, vorticity)[79,85–87]:

$$\rho = \frac{1}{2} \epsilon_{no} \mathbf{m} \cdot \left[\partial_n \mathbf{m} \times \partial_o \mathbf{m}\right],$$

with $n,o = \{x,y\}$. This is widely used for 2D topological nontrivial magnetic distributions, e.g., solitons[88], vortices[89] and skyrmions[64], in planar magnets and thin-film shells.

## Sample preparation
A nominally 50-nm-thick permalloy layer is deposited on top of 80 nm polystyrene spheres by means of magnetron sputtering at room temperature (base pressure $10^{-8}$ mbar, Ar is used as a sputter gas at a pressure of $10^{-3}$ mbar). Before the deposition, polystyrene spheres were distributed from solvent across a copper TEM grid with supporting lacey carbon film.

## Bright-field TEM tomography
A tilt series of bright-field Lorentz TEM measurements within an angular tilt range of $\pm 68°$ in $4°$ steps is acquired for a representative permalloy cap on top of a polystyrene sphere attached to the bar of the lacey carbon support covered with permalloy. The bright-field TEM images of the tilt series are then converted to projected attenuation maps by computing the negative logarithm. Coarse displacements between successive projections are corrected by cross-correlation, whereas the fine alignment, i.e., the accurate determination of the tilt axis and correction for sub-pixel displacements, is conducted by a self-implemented center-of-mass method and common line approach. The tomographic reconstruction of the aligned tilt series is carried out using the weighted simultaneous iterative reconstruction technique (W-SIRT)[54] using five iterations. The in-house developed W-SIRT method utilizes at each SIRT iteration a weighted instead of a simple back projection as in the case of conventional SIRT. This improves the convergence properties of the tomographic reconstructions. The obtained tomogram with a spatial resolution of about 0.5 nm reveals a corrugated surface of the object (Supplementary Movie 1 and Supplementary Note 6A for details). The latter is both a real property of the permalloy layer deposited by sputtering and a reconstruction artefact due to noise and local dynamical scattering contrast if single grains of the poly-crystalline permalloy layer are in low-index zone axis orientation with respect to the electron beam. Moreover, due to the incomplete tomographic tilt range ($\pm 68°$ instead of $\pm 90°$), the lateral resolution in the directions, which are not covered by the tilt series, is reduced. For these reasons, the tomogram was smoothed (i.e., regularized), before the surface of the permalloy cap is finally extracted as input for the micromagnetic simulations.

## Electron holography
Electron holograms of polystyrene spheres covered with permalloy caps were acquired using a double-corrected FEI Titan³ 80 – 300 microscope (ThermoFisher Comp., USA) operated in imaging-corrected Lorentz mode (conventional objective lens switched off) at an acceleration voltage of 300 kV. The voltage of the electrostatic Möllenstedt bisprism is set to 210 V yielding a hologram

fringe spacing of 1.7 nm and fringe contrast of 15 % acquired with a 2k by 2k Gatan Ultracan CCD camera. Amplitude and phase images were reconstructed from the electron holograms by Fourier techniques incorporating empty holograms for correction of imaging artifacts, such as distortions of the camera and projective lenses[90]. The reconstructed phase images contain the phase shift between object wave and unperturbed reference wave that can be expressed by

$$\varphi(x,y) = \int_{-\infty}^{\infty} dz \left[ C_E V(x,y,z) - \frac{e}{\hbar} A_z(x,y,z) \right], \qquad (5)$$

where $C_E$ is an interaction constant depending on the electron beam energy, $V(x,y,z)$ the three-dimensional electrostatic object potential, $e$ the electron charge, $\hbar$ the reduced Planck constant, and $A_z(x,y,z)$ the component of the magnetic vector potential parallel to the electron beam direction $z$. Accordingly, the first term can be considered as electric phase shift $\varphi_{el}$ and the second as magnetic one $\varphi_{mag}$. The latter can also be expressed by the magnetic flux from which the projections of the lateral components $B_x$ and $B_y$ of the magnetic flux density (induction) $\mathbf{B} = \{B_x, B_y, B_z\}$ may be obtained by differentiating $\varphi_{mag}$ according to

$$\nabla_{x,y} \varphi_{mag}(x,y) = \frac{e}{\hbar} \int_{-\infty}^{\infty} dz \begin{Bmatrix} B_y(x,y,z) \\ -B_x(x,y,z) \end{Bmatrix}. \qquad (6)$$

To separate electric and magnetic phase shift, for each phase image a second phase image of the specimen flipped up-side down is acquired. Due to the odd time-reversal symmetry of the magnetic induction, the magnetic phase shift changes its sign in the second phase, hence, can be calculated by half of the difference between phase images before and after flipping. Prior to the separation, the phase images were firstly corrected from residual image aberrations (e.g., as described in ref. [90]) and secondly aligned by affine image registration methods to minimize artifacts in the magnetic phase shifts, especially towards the boundaries of the nanocap. Exploiting relation (6), the projected magnetic induction components $B_x$ and $B_y$ are finally computed. To establish different remanent vortex (antivortex) states of the nanocaps (Figs. 4 and 5), we applied an out-of-plane magnetic field using the objective lens. We started with the application of +1500 mT resulting in the magnetic state with $C = +1$ and $P = -1$ (Fig. 5), followed by −35 mT resulting in $C = -1$ and $P = -1$ (Fig. 4), followed by −400 mT resulting in $C = -1$ and $P = +1$ (Fig. 5), and finally, +400 mT resulting in $C = +1$ and $P = +1$ (Fig. 4).

## Data availability

All data that support the plots within this paper and other findings of this study are available from the corresponding authors upon reasonable request. Source data are provided with this paper.

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

## Acknowledgements

This work is financed in part via the German Research Foundation (DFG) under Grants No. MA 5144/14-1, MA 5144/22-1, MA 5144/24-1, KA 5069/1-1, KA 5069/3-1, VO 2598/1-1 and MC 9/22-1. D.W. and A.L. have received funding from the European Research Council (ERC) under the Horizon 2020 research and innovation program of the European Union (grant agreement number 715620). B.B. received funding from the Würzburg-Dresden Cluster of Excellence on Complexity and Topology in Quantum Matter-ct.qmat (EXC 2147, project-id 390858490). A.K. acknowledges Prof. István Kézsmárki for helpful discussions on symmetries.

## Author contributions

D.W. and A.L. performed the imaging of asymmetric cap structures, including 3D shape reconstruction and magnetic texture identification. O.M.V., O.V.P., D.D.Sh., and D.M. formulated the theoretical problem. O.M.V., O.V.P., and A.K. performed full-scale micromagnetic simulations. O.M.V. carried out a detailed analysis of the experimental and micromagnetic results with the support of O.V.P, D.W., A.L., and D.M. The manuscript was written by O.M.V., O.V.P., and D.M. with contribution from A.K., D.D.Sh., B.B., J.F., and A.L.

## Funding

## Competing interests

The authors declare no competing interests.
