## [Peer Review File · Nature Communications]

Reviewers' Comments:

Reviewer #1:

Remarks to the Author:

This is a comprehensive study of the vortex string in asymmetric nanodisks. Due to the bump and its induced inversion symmetry breaking, the magnetization effectively feels a off-diagonal spin interaction, and results in chiral structures. The vortex string thus has its circularity and polarity locked, similar as the Dzyaloshinskii-Moriya interaction induced skyrmions. Furthermore, the bending of the vortex string is related to the off-center position of the bump. Their simulation results are further supported by the electron holography experiment.

The results of this article are reasonable; maybe too reasonable that most results are quite expected. However, given the heavy calculation and completeness of the study, it is still worth publishing in Nature Communications. The presentation is also superb to attract general audience. I therefore recommend its publication as is.

Reviewer #2:

Remarks to the Author:

The authors studied an intriguing issue of the nonlocal chiral symmetry breaking in complex curvilinear objects. Although many factors are involved in stabilizing topological magnetic structures under geometrical constraint, the authors have clearly elucidated an essential role of the geometrical symmetry of the host in shaping the vortex strings. Supporting stuffs are helpful to resolve the multivariate and intricate phenomena and the findings are quite informative; thus I recommend to be published.

My minor concerns are as follows:

1) In Fig2b,c,d,g; Why the different values of iso- Ω contours are selected? So as to compare the width of the vortex, the same Ω value should be selected.

2) In Fig.2d; Another local maximum can be seen near the bump (opposite side of the $\Omega=0.75$ string). Does this mean the branching of the vortex?

Reviewer #3:

Remarks to the Author:

In the manuscript entitled "Chirality coupling in topological magnetic textures with multiple magnetochiral parameters", the authors theoretically investigated the deformation manner of vortex string in ferromagnetic films with different top and bottom surface shapes. Their analysis suggests that the bending manner (torsion) of the vortex string should be coupled with the helicity of magnetic texture in such asymmetric systems, and this prediction was partly supported by the associated experiments for ferromagnetic permalloy nanocap deposited on spherical non-magnetic objects.

Three-dimensional string-like spin textures, such as vortex string, skyrmion tube, and hopfion, are attracting attention as unique examples of topologically protected objects in magnetic materials. Their direct experimental visualization has been achieved only very recently, and the present work may provide a new general guideline to understand the bending manner of such string-like magnetic objects in asymmetric systems. However, I also think that the manuscript has several problems as discussed below, and would like the authors to answer the following questions and comments before making a firm recommendation.

1. From the viewpoint of symmetry, two possible states with the same magnetic helicity (Fig. 4) can be converted into each other by time-reversal operation, and therefore their vortex strings must show the same manner of bending. On the other hand, the states with different helicity cannot be converted into each other by any symmetry operation, for such an asymmetric system as employed in the present experiments. In this context, the appearance of the same (different) torsion for the magnetic states with the same (different) helicity is somewhat predictable. What is the importance of this finding? Can the authors derive any further conclusions beyond the simple symmetry analysis?

2. In the present experiment, the authors confirmed that the bending manner (torsion) of the vortex string is coupled with the helicity (=CP) of magnetic texture. Despite its relevance to the main conclusion, this data was provided only in Supplementary Figs. 15-18. I think that the observed string shape and its torsion profile for each configuration should be provided in the main text.

3. Have the authors investigated different shape/symmetry of ferromagnetic nanocaps to fully confirm their prediction in Fig. 6?

4. The manuscript describes a complicated story in a complicated manner, and I naively feel that it would be very difficult for general readers (in particular the ones not familiar with this kind of topic) to understand the essence of the present work. This is partly because the authors use many letter symbols in the figures but their corresponding meaning/definition are not visually illustrated. I recommend the authors to provide more illustrations in their figures in the main text that support the intuitive explanation of the meaning of each letter symbol.

5. I found many typos associated with figure numbers, which should be carefully corrected.

Response letter

We thank the Referees for providing their constructive remarks that helped us improving clarity of the manuscript. Our itemized responses to the remarks of the Referees are listed below with all changes in the manuscript that are indicated in blue.

Referee #1

Comment 1:

This is a comprehensive study of the vortex string in asymmetric nanodisks. Due to the bump and its induced inversion symmetry breaking, the magnetization effectively feels a off-diagonal spin interaction, and results in chiral structures. The vortex string thus has its circularity and polarity locked, similar as the Dzyaloshinskii-Moriya interaction induced skyrmions. Furthermore, the bending of the vortex string is related to the off-center position of the bump. Their simulation results are further supported by the electron holography experiments.

The results of the this article are reasonable; maybe too reasonable that most results are quite expected. However, given the heavy calculation and completeness of the study, it is still worth publishing in Nature Communications. The presentation is also superb to attract general audience. I therefore recommend its publication as it is.

Answer:

We thank the Referee for his/her positive evaluation of our work and recommendation for its publication in Nature Communications.

Following the remark of the Referee, we put more emphasis on the fundamental novelty of the work. In particular, we note that the reported here (i) helix-shaped distortion of the vortex string and (ii) link between the helix torsion and the vortex helicity are new effects which were not reported before.

The following text is added in the discussion section of the manuscript (page 9):

Deformations of a texture should not be necessarily expected as a homochiral or unidirectional in a general sense. Namely, microwave excitations of chiral skyrmion tubes demonstrate both sense of left- and right-handed rotations independently of the skyrmion chirality.^{1,2} This is in contrast to our finding, that even at equilibrium and in the absence of the Dzyaloshinskii–Moriya interaction, the vortex string not only bends into helix, but also the helix chirality is unique for the given vortex symmetry. It should be noted, that the similar curling deformation of the same chirality has been recently reported for skyrmion tubes in the asymmetrically shaped helimagnetic samples.³ Thus, we anticipate that our findings are generic for a wide class of magnetic nanotextures and provide an explanation of their structure in geometrically asymmetric samples.

Referee #2

Comment 1:

The authors studied an intriguing issue of the nonlocal chiral symmetry breaking in complex curvilinear objects. Although many factors are involved in stabilizing topological magnetic structures under geometrical constraint, the authors have clearly elucidated an essential role of the geometrical symmetry of the host in shaping the vortex strings. Supporting stuffs are helpful to resolve the multivariate and intricate phenomena and the findings are quite informative; thus I recommend to be published.

Answer:

We thank the Referee for the positive assessment of our work and recommendation to publish it in Nature Communications. We revised the manuscript accordingly to the remarks of the Referee.

Comment 2:

My minor concerns are as follows: 1) In Fig. 2b,c,d,g; Why the different values of iso- Ω contours are selected? So as to compare the width of the vortex, the same Ω values should be selected.

Answer:

We thank the Referee for this remark. The construction of the presented iso-surfaces for the normalized topologically charge density $\tilde{\Omega}$ is derived from the Ω , which is based on the calculation of magnetization spatial derivatives. The latter reach their maxima near the top and bottom surfaces as well as in close proximity to the vortex core. Thus, as absolute values of Ω distributions for thin and thick magnetic nanodots are quite different due to the influence of volume magnetostatic charges, the corresponding $\tilde{\Omega}$ varies accordingly. To emphasize this aspect, we replace Fig. 2 with the new one (shown below Fig. R1), where panels (c), (d) and (g) contain also iso-surface $\tilde{\Omega} = 0.6$.

Comment 3:

2) In Fig. 2d; Another local maximum can be seen near the bump (opposite side of the $\Omega = 0.75$ sting). Does this mean the branching of the vortex?

Answer:

The confusion might be caused by the color scheme, which indicates the strength of the surface magnetostatic charges. To avoid misinterpretations, we changed the color scheme in Fig. 2.

We added the following remark to the manuscript (page 3):

Results of the micromagnetic simulations for any of the states shown in Fig. R1a reveal that bending of the vortex string is not accompanied by its branching. This is in contrast to the reported bifurcation of skyrmion tubes into a complex 3D texture.⁴⁻⁶

Referee #3

Comment 1:

In the manuscript entitled “Chirality coupling in topological magnetic textures with multiple magneto-chiral parameters”, the authors theoretically investigated the deformation manner of vortex string in ferromagnetic films with different top and bottom surface shapes. Their analysis suggests that the bending manner (torsion) of the vortex string should be coupled with the helicity of magnetic texture in such asymmetric systems, and this prediction was partly supported by the associated experiments for ferromagnetic permalloy nanocap deposited on spherical non-magnetic objects.

Three-dimensional string-like spin textures, such as vortex string, skyrmion tube, and hopfion, are attracting attention as unique examples of topologically protected objects in magnetic materials. Their direct experimental visualization has been achieved only very recently, and the present work may provide a new general guideline to understand the bending manner of such string-like magnetic objects in asymmetric systems. However, I also think that the manuscript has several problems as discussed below, and would like the authors to answer the following questions and comments before making a firm recommendation.

Answer:

We thank the Referee #3 for his/her remarks, which we used to improve the manuscript. Our itemized responses are listed below.

Comment 2:

From the viewpoint of symmetry, two possible states with the same magnetic helicity (Fig. 4) can be converted into each other by time-reversal operation, and therefore their vortex strings must show the same manner of bending. On the other hand, the states with different helicity cannot be converted into each other by any symmetry operation, for such an asymmetric system as employed in the present experiments. In this context, the appearance of the same (different) torsion for the magnetic states with the same (different) helicity is somewhat predictable. What is the importance of this finding?

Answer:

We would like to emphasize that the key novelty of the present manuscript is the appearance and interconnection of two magneto-chiral properties in asymmetric magnetic nanodots: (i) magnetic helicity of the vortex, \mathcal{C} , that is invariant with respect to the time-reversal operation and (ii) chirality of the vortex string curling deformation. Namely, the latter appears to be only right-handed for $\mathcal{C} = +1$ and left-handed for $\mathcal{C} = -1$. We demonstrated experimentally, that this interconnection is so strong that switching of the vortex polarity introduces the appearance of an additional antivortex that together with the initial vortex form a topologically trivial state, see Fig. 5.

Comment 3:

Can the authors derive any further conclusions beyond the simple symmetry analysis?

Answer:

The study of nonlocal chiral symmetry breaking effects in asymmetric curvilinear geometries beyond the symmetry analysis requires a general theory of asymmetric media, which in general case of arbitrary shapes could not be constructed. Still, the extension of the existing theoretical framework⁷ from thin curvilinear shells to thicker asymmetrically curved geometries is an appealing future research direction, which should be addressed in a separate work.

(4) In the present experiment, the authors confirmed that the bending manner (torsion) of the vortex string is coupled with the helicity (= CP) of magnetic texture. Despite its relevance to the main conclusion, this data was provided only in Supplementary Figs. 15-18. I think that the observed string shape and its torsion profile for each configuration should be provided in the main text.

Answer:

Following this suggestion of the Referee, we introduced a new Fig. 7 (Fig. R2 shown below) to the Results section of the manuscript.

(5) Have the authors investigated different shape/symmetry of ferromagnetic nanocaps to fully confirm their prediction in Fig. 6?

Answer:

We thank the Referee for this comment. In the present manuscript we focused our investigation on the nonlocal chiral symmetry breaking effect in thick asymmetric Py caps. Arguably, this is the most natural and straightforward choice of geometry as it allows to stabilize a vortex texture and provides an asymmetry of the top and bottom surfaces. To be sure that the results are reproducible, we performed magnetic imaging of two distinct cap structures. In particular, data shown in Fig. 3 concern one cap structure and data shown in Figs. 4 and 5 concern another cap structure. Although the caps are not exactly the same, we do not observe qualitative differences in their behaviour. In the manuscript we investigated two different Py cap geometries. The first one was used to perform structural and magnetic investigation of the as prepared sample, while the second one was used to investigate the evolution of the vortex state upon switching.

The following changes were introduced to the text to clarify this point (page 6):

To understand the formation of magnetochiral vortex states, we performed additional experimental investigations of another asymmetric nanocap possessing vortices of different circulation and polarity.

(6) The manuscript describes a complicated story in a complicated manner, and I naively feel that it would be very difficult for general readers (in particular the ones not familiar with this kind of topic) to understand the essence of the present work. This is partly because the authors use many letter symbols in the figures but their corresponding meaning/definition are not visually illustrated. I recommend the authors to provide more illustrations in their figures in the main text that support the intuitive explanation of the meaning of each letter symbol.

Answer:

Following the remark of the Referee, we added an additional Fig. 7 to the main text. We reduced the number of letter symbols to the necessary minimum, which is needed to describe physics of the curved magnetic vortex string. Most of these symbols are related to geometrical parameters like curvature, torsion, radius, pitch, thickness. These parameters are indicated in figure panels. More unusual parameter is the normalized topologically charge density $\tilde{\Omega}$. But it is also shown with respective iso-surfaces in figures. Vortex properties including polarity and circulation are also introduced in Fig. 1. Furthermore, in addition to the main text, further details on the introduced mathematical symbols and their physical meaning are provided in supporting information.

(7) *I found many typos associated with figure numbers, which should be carefully corrected.*

Answer:

We carefully checked the whole manuscript and corrected all spotted typos.

References

1. Kravchuk, V. P., Röbber, U. K., van den Brink, J. & Garst, M. Solitary wave excitations of skyrmion strings in chiral magnets. *Physical Review B* **102** (2020).
2. Seki, S. *et al.* Propagation dynamics of spin excitations along skyrmion strings. *Nature Communications* **11** (2020).
3. Wolf, D. *et al.* Unveiling the three-dimensional magnetic texture of skyrmion tubes. *Nature Nanotechnology* **17**, 250–255 (2022). URL <http://dx.doi.org/10.1038/s41565-021-01031-x>.
4. Milde, P. *et al.* Unwinding of a skyrmion lattice by magnetic monopoles. *Science* **340**, 1076–1080 (2013). URL <http://dx.doi.org/10.1126/science.1234657>.
5. Birch, M. T. *et al.* Real-space imaging of confined magnetic skyrmion tubes. *Nature Communications* **11**, 1726 (2020). URL <http://dx.doi.org/10.1038/s41467-020-15474-8>.
6. Xia, J. *et al.* Bifurcation of a topological skyrmion string. *Physical Review B* **105**, 214402 (2022). URL <http://dx.doi.org/10.1103/physrevb.105.214402>.
7. Sheka, D. D. *et al.* Nonlocal chiral symmetry breaking in curvilinear magnetic shells. *Communications Physics* **3**, 128 (2020). URL <https://doi.org/10.1038/s42005-020-0387-2>.

Figure R1. Vortex states in nanodisks of different geometry. (a) Product of the total surface and volume magnetostatic charges Σ and Λ , respectively, for different nanodisk geometries. Symbols correspond to the results of full-scale micromagnetic simulations of nanodisks with a radius of 150 nm. States, which are typical for the regions marked by different symbols, are shown in the following panels. (b) The central part of the symmetric nanodisk (thickness: $h = 30$ nm) contains the straight vortex line. These states in symmetric nanodisks are marked by circles in (a). Here and below, the red sandglass-like region corresponds to the distribution of the normalized topological charge density, $\tilde{\Omega} = 0.6$, and determines the spatial localization and shape of the vortex string. Isosurfaces for other $\tilde{\Omega}$ are shown to illustrate the change of the vortex core width profile over the sample thickness. (c) The central part of the highly asymmetric nanodisk of small thickness ($h = 5$ nm), accommodating a tall off-centered Gaussian bump ($t = 50$ nm, $b = 20$ nm and $x_0 = 10$ nm) at its top surface, contains a pinned vortex string with a very small curvature. These states are marked by diamonds in (a). Here, $\langle \kappa_v \rangle = 0.7 \mu\text{m}^{-1}$, which results in a substantial torsion $\langle \tau_v \rangle = 3.7 \mu\text{m}^{-1}$ (helix with radius $\langle R_v \rangle = 9.1\ell$ and pitch $\langle P_v \rangle = 308.2\ell$). (d) The central part of an asymmetric nanodisk ($h = 30$ nm) with a shallower Gaussian bump ($t = 20$ nm, width $b = 10$ nm and shift $x_0 = 10$ nm) on the top surface of the disk contains a short vortex string, which is expelled from the disk center. These states are marked by triangles in (a). Here, the average string curvature $\langle \kappa_v \rangle = 6.7 \mu\text{m}^{-1}$ and torsion $\langle \tau_v \rangle = 3 \mu\text{m}^{-1}$ correspond to the helix with a radius $\langle R_v \rangle = 24.1\ell$ and pitch $\langle P_v \rangle = 60\ell$. Distribution of (e) curvature κ_v and (f) torsion τ_v along the vortex string shown in (d). Symbols indicate on-site values, solid lines represent trends and dashed lines show mean values. (g) The central part of a thick asymmetric nanodisk with a Gaussian bump ($h = 30$ nm, $t = 40$ nm, $b = 20$ nm and $x_0 = 10$ nm) containing a curled vortex string (c.f. Fig. 1b). These states are marked by squares in (a). Here, the average curvature $\langle \kappa_v \rangle = 4 \mu\text{m}^{-1}$ and torsion $\langle \tau_v \rangle = 3.3 \mu\text{m}^{-1}$ correspond to the helix with a radius $\langle R_v \rangle = 27.6\ell$ and pitch $\langle P_v \rangle = 144.7\ell$. Distribution of (h) curvature κ_v and (i) torsion τ_v along the vortex string shown in (g). Symbols indicate on-site values, solid lines represent trends and dashed lines show mean values.

Figure R2. Vortex states in a truncated nanocap studied experimentally. Panels (a), (f), (j) and (n) represent the reconstructed vortex lines (black tubes) inside the truncated asymmetric nanocap with vortices of different magnetic helicity: (a) for $P = +1$, $C = +1$; (f) for $P = -1$, $C = -1$; (j) for $P = +1$, $C = -1$; (n) for $P = -1$, $C = -1$. Their rescaled shapes extracted from the calculation of $\tilde{\Omega}$ are shown on (b), (g), (k) and (o), respectively. The resulting distributions of vortex string curvature and torsion are shown in (d), (h), (l), (p) and (e), (i), (m), (q), respectively. All isosurfaces are constructed for $\tilde{\Omega} = 0.35$.

Reviewers' Comments:

Reviewer #3:

Remarks to the Author:

In the revised manuscript, the authors appropriately addressed all the issues raised in the previous communication. I have no further comment, and recommend the publication of the manuscript in Nature Communications.